# Comparison of First- and Second-Generation Drug-Eluting Stents in Patients with ST-Segment Elevation Myocardial Infarction Based on Pre-Percutaneous Coronary Intervention Thrombolysis in Myocardial Infarction Flow Grade

**DOI:** 10.3390/jcm10020367

**Published:** 2021-01-19

**Authors:** Yong Hoon Kim, Ae-Young Her, Myung Ho Jeong, Byeong-Keuk Kim, Sung-Jin Hong, Seunghwan Kim, Chul-Min Ahn, Jung-Sun Kim, Young-Guk Ko, Donghoon Choi, Myeong-Ki Hong, Yangsoo Jang

**Affiliations:** 1Department of Internal Medicine, Division of Cardiology, Kangwon National University School of Medicine, 156 Baengnyeong Road, Chuncheon 24289, Korea; hermartha1@gmail.com; 2Cardiovascular Center, Department of Cardiology, Chonnam National University Hospital, Gwangju 61469, Korea; myungho@chollian.net; 3Division of Cardiology, Severance Cardiovascular Hospital, Yonsei University College of Medicine, Seoul 03722, Korea; kimbk@yuhs.ac (B.-K.K.); HONGS@yuhs.ac (S.-J.H.); DRCELLO@yuhs.ac (C.-M.A.); kjs1218@yuhs.ac (J.-S.K.); ygko@yuhs.ac (Y.-G.K.); cdhlyj@yuhs.ac (D.C.); mkhong61@yuhs.ac (M.-K.H.); jangys1212@yuhs.ac (Y.J.); 4Division of Cardiology, Inje University College of Medicine, Haeundae Paik Hospital, Busan 48108, Korea; cloudksh@gmail.com

**Keywords:** ST-segment elevation myocardial infarction, percutaneous coronary intervention, reperfusion, stents

## Abstract

This study aims to investigate the two-year clinical outcomes between first-generation (1G) and second-generation (2G) drug-eluting stents (DES) based on pre-percutaneous coronary intervention (PCI) thrombolysis in myocardial infarction (TIMI) flow grade (pre-TIMI) in patients with ST-segment elevation myocardial infarction (STEMI). Overall, 17,891 STEMI patients were classified into two groups: pre-TIMI 0/1 group (*n* = 12,862; 1G-DES (*n* = 4318), 2G-DES (*n* = 8544)) and pre-TIMI 2/3 group (*n* = 5029; 1G-DES (*n* = 2046), 2G-DES (*n* = 2983)). During a two-year follow-up period, major adverse cardiac events (MACEs) defined as all-cause death, recurrent myocardial infarction (re-MI), or any repeat revascularization and stent thrombosis (ST) were considered as the primary and the secondary outcomes. In the pre-TIMI 0/1 and 2/3 groups, the cumulative incidences of MACEs (adjusted hazard ratio (aHR): 1.348, *p* < 0.001, and aHR: 1.415, *p* = 0.02, respectively) and any repeat revascularization (aHR: 1.938, *p* < 0.001, and aHR: 1.674, *p* = 0.001, respectively) were significantly higher in the 1G-DES than in the 2G-DES. However, sirolimus-eluting stent showed similar cumulative incidence of any repeat revascularization compared with zotarolimus-eluting stent and biolimus-eluting stent in both pre-TIMI 0/1 and 2/3 groups. The cumulative incidences of all-cause death, re-MI, and ST were similar between the 1G-DES and 2G-DES groups. In this study, 2G-DES showed better clinical outcomes than 1G-DES concerning MACEs and any repeat revascularization regardless of pre-TIMI. However, more research is needed to support these results.

## 1. Introduction

Although second-generation drug-eluting stents (2G-DES) have a newly advanced biocompatible polymer compared to first-generation (1G-DES), their comparative results are conflicting [1,2,3]. Previous studies [4,5] demonstrated that pre-percutaneous coronary intervention (pre-PCI) thrombolysis in myocardial infarction (TIMI) flow grade (pre-TIMI) was a significant predictor of survival in ST-segment elevation myocardial infarction (STEMI) patients. If the blood supply goes down, remnant oxygen in the ischemic area of the myocardium is disappeared within seconds. Therefore, after a certain duration of complete ischemia, there is no treatment strategy that can salvage ischemic myocardium [6]. However, cardiomyocytes that are exposed to low residual oxygen levels may be able to maintain sufficient adenosine triphosphate to survive for an extended period, even if the adenosine triphosphate is insufficient to enable their contraction [6]. Hence, pre-TIMI 0/1 and pre-TIMI 2/3 [7] are apparently different milieu in patients with STEMI. Recently, Yildiz et al. [8] suggested that the pre-TIMI was closely linked with percentage of patients getting PCI, the use of glycoprotein IIb/IIIa, and stent type. Therefore, pre-TIMI can be another significant variable for comparing major clinical outcomes between 1G-DES and 2G-DES. However, the majority of previous such comparative studies between these two different generation DES were not based on pre-TIMI in patients with STEMI [9,10]. Here, we compared the two-year major clinical outcomes between 1G-DES and 2G-DES based on these two different pre-TIMIs of the culprit coronary artery in patients with STEMI.

## 2. Materials

### 2.1. Study Design and Population

From the Korea acute myocardial infarction (AMI) registry (KAMIR) [11], 24,549 STEMI patients who received successful stent implantation between November 2005 and June 2015 were evaluated. The characteristics of the KAMIR are already published [11]. Among them, those with incomplete laboratory results (*n* = 4004 (16.3%)), who were lost to follow-up (*n* = 994 (4.0%)), who received bare-metal stent (*n* = 1439 (5.9%)), or who received 1G-DES and 2G-DES concomitantly (*n* = 221 (0.9%)) were excluded. Overall, 17,891 STEMI patients were classified into two groups: pre-TIMI 0/1 group (*n* = 12,862 (71.9%)) and pre-TIMI 2/3 group (*n* = 5029 (28.1%)). Thereafter, these two groups were subdivided into patients who received 1G-DES (Group A1 (*n* = 4318 (33.6%)) and Group A2 (*n* = 2046 (40.7%))) and those who received 2G-DES (Group B1 (*n* = 8544 (67.4%)) and Group B2 (*n* = 2983 (59.3%))) (Figure 1). All patients provided written informed consent before enrollment. During the follow-up period, any adverse events of the enrolled patients were carefully monitored at the outpatient clinic, by phone calls, or by reviewing their charts. Altogether, 17,891 STEMI patients completed the scheduled follow-up. Additionally, an independent event-adjudicating committee evaluated all clinical events. The event adjudication process was mentioned in a previous publication [11]. This study protocol was approved by the ethics committee at each participating center and the Chonnam National University Hospital Institutional Review Board (IRB) ethics committee (CNUH-2011-172) according to the ethical guidelines of the 1975 Declaration of Helsinki.

### 2.2. Percutaneous Coronary Intervention (PCI) Procedure and Medical Treatment

We performed diagnostic coronary angiography (CAG) and PCI according to the standard techniques [12]. We loaded the loading doses of antiplatelet agents as follows: aspirin 200–300 mg, clopidogrel 300–600 mg, and ticagrelor 180 mg or prasugrel 60 mg. Dual antiplatelet therapy (DAPT; combination of aspirin 100 mg/day and clopidogrel 75 mg/day or ticagrelor 90 mg twice daily or prasugrel 5–10 mg/day) was recommended for at least 12 months. The individual operators were free to choose DAPT or triple antiplatelet therapy (aspirin + clopidogrel + cilostazol (100 mg twice daily)) during the operation.

### 2.3. Study Definitions and Clinical Outcomes

We defined STEMI according to the current guidelines [13,14]. The degree of pre-TIMI was assessed by the investigators [15]. During a two-year follow-up period, major adverse cardiac events (MACEs) defined as all-cause death, recurrent myocardial infarction (re-MI), or any repeat revascularization and stent thrombosis (ST) were considered as the primary and the secondary outcomes. All-cause death consisted of cardiac death (CD) or non-CD. Target lesion revascularization (TLR), target vessel revascularization (TVR), and non-TVR were included in any repeat revascularization. The definitions of successful PCI, complete revascularization, incomplete revascularization, re-MI, TLR, TVR, and non-TVR were as previously reported [16,17].

### 2.4. Statistical Analysis

Regarding continuous variables, we compared differences between groups using the unpaired t-test. Data are expressed as mean ± standard deviation (SD) or as median (quartiles 1–3). Regarding discrete variables, we compared intergroup differences using the chi-square test or Fisher’s exact test. Data are expressed as counts and percentages. Various clinical outcomes were estimated using Kaplan–Meier curve analysis, and intergroup differences were compared using the log-rank test. Two-tailed *p*-values < 0.05 were considered statistically significant. We tested all variables with *p* < 0.1 in the univariate analysis. After univariate analysis, we tested all variables with *p* < 0.05 in the multivariate analysis. These variables included the following: male, age, left ventricular ejection fraction (LVEF), Killip class III/IV, systolic blood pressure, cardiogenic shock, cardiopulmonary resuscitation (CRP) on admission, hypertension, diabetes mellitus (DM), dyslipidemia, prior PCI, current smoker, peak creatine kinase myocardial band (CK-MB), peak troponin-I, N-terminal pro-brain natriuretic peptide (NT-ProBNP), serum creatinine, total cholesterol, triglyceride, high-density lipoprotein-cholesterol, low-density lipoprotein-cholesterol, clopidogrel, ticagrelor, prasugrel, cilostazole, beta-blocker (BB), calcium channel blocker, single-vessel disease, more than three diseased vessels, left anterior descending coronary artery (infarct-related artery (IRA) and treated vessel), right coronary artery (IRA), American College of Cardiology/American Heart Association (ACC/AHA) type B1/B2/C lesion, in-hospital glycoprotein IIb/IIIa inhibitor, intravascular ultrasound (IVUS), culprit-only PCI, stent diameter, stent length, and number of deployed stents. Additionally, we evaluated the cumulative incidence of any repeat revascularization according to deployed stent type (Appendix A) using analysis of variance (ANOVA) or the Jonckheere-Terpstra test. Because there are some issues of multiple comparisons and concern for false-positive association during the analysis, we adjusted significance thresholds for each comparison by using the Benjamini-Hochberg false discovery control procedure with a set false discovery rate of 5% during a post-hoc analysis. Regarding categorical variables, we compared intergroup differences using the chi-square test or Fisher’s exact test. All statistical analyses were performed using SPSS software version 20 (IBM, Armonk, NY, USA).

## 3. Results

### 3.1. Baseline Characteristics

Table 1 shows the baseline characteristics of the study population. In the pre-TIMI 0/1 (Groups A1 and B1) and 2/3 (Groups A2 and B2) groups, the mean LVEF was preserved (≥50%). Significantly more patients in the pre-TIMI 0/1 group than in the pre-TIMI 2/3 group received in-hospital glycoprotein IIb/IIIa. The prescription rates of clopidogrel, cilostazole, and calcium channel blockers as the discharge medications, more than three diseased vessels, ACC/AHA type B1 lesion, multivessel PCI, number of deployed stents, and mean serum creatinine and high-density lipoprotein-cholesterol levels were significantly higher in the 1G-DES (Groups A1 and A2) groups than in the 2G-DES (Groups B1 and B2) groups. In contrast, the number of men, number of patients requiring CPR on admission, with dyslipidemia or previous PCI, prescription rates of ticagrelor, prasugrel, and BBs as the discharge medications, patients with single-vessel disease, ACC/AHA type B2 lesions, and culprit-only PCI, number of patients requiring IVUS, mean triglyceride level, and mean deployed stent diameter were significantly higher in the 2G-DES groups than in the 1G-DES groups. However, the mean door-to-balloon time and number of complete revascularizations in case of multivessel PCI, and number of patients with cardiogenic shock, hypertension, and diabetes mellitus, and the number who were current smokers, were similar between the two groups.

### 3.2. Clinical Outcomes

The cumulative incidence of major clinical outcomes during the two-year follow-up period is shown in Table 2 and Table 3, and Figure 2 and Figure 3. Appendix A shows the independent predictors for MACEs and any repeat revascularization in the total study population at 2 years. In the total study population, after adjustment, the cumulative incidence of MACEs (adjusted hazard ratio (aHR): 1.210; 95% confidence interval (CI): 1.057–1.398; *p* = 0.005), all-cause death (aHR: 1.350; 95% CI: 1.095–1.699; *p* = 0.006), and CD (aHR: 1.498; 95% CI: 1.180–1.910; *p* = 0.001) were significantly higher in the pre-TIMI 0/1 group than in the pre-TIMI 2/3 group. The cumulative incidence of MACEs (aHR: 1.233; 95% CI: 1.095–1.388; *p* < 0.001) and any repeat revascularization (aHR: 1.832; 95% CI: 1.528–2.197; *p* < 0.001) were significantly higher in the 1G-DES (Groups A1 and A2) than in the 2G-DES (Groups B1 and B2) groups.

In the pre-TIMI 0/1 group, after adjustment, the cumulative incidence of MACEs (aHR: 1.348; 95% CI: 1.84–1.536; *p* < 0.001) and any repeat revascularization (aHR: 1.938; 95% CI: 1.605–2.340; *p* < 0.001) were significantly higher in the 1G-DES groups than in the 2G-DES groups. Similarly, in the pre-TIMI 2/3 group, after adjustment, the cumulative incidence of MACEs (aHR: 1.415; 95% CI: 1.142–1.953; *p* = 0.002) and any repeat revascularization (aHR: 1.674; 95% CI: 1.244–2.352; *p* = 0.001) were also significantly higher in the 1G-DES groups than in the 2G-DES groups. However, in the pre-TIMI 0/1 and 2/3 groups, the cumulative incidence of all-cause death, CD, re-MI, and ST were similar between the 1G-DES and 2G-DES groups.

Figure 3 and Table 4 show the cumulative incidence of any repeat revascularization according to deployed stent type. Moreover, Appendix A show the baseline characteristics according to deployed stent type. Paclitaxel-eluting stents (PES) showed a higher cumulative incidence of any repeat revascularization than sirolimus-eluting stents (SES) in the total study population (aHR: 1.415; 95% CI; 1.132–1.769; *p* = 0.002), pre-TIMI 0/1 (aHR: 1.374; 95% CI; 1.049–1.798; *p* = 0.021), and pre-TIMI 2/3 (aHR: 1.533; 95% CI; 1.035–2.330; *p* = 0.034) groups. However, sirolimus-eluting stent showed similar cumulative incidence of any repeat revascularization compared with zotarolimus-eluting stent and biolimus-eluting stent in both pre-TIMI 0/1 and 2/3 groups. Appendix A shows the independent predictors for MACEs and any repeat revascularization in the total study population at 2 years. Old age (≥65 years), cardiogenic shock, CPR on admission, diabetes mellitus, BB, left main coronary artery as a treated vessel, single-vessel disease, more than three diseased vessels, and culprit-only PCI were significant independent predictors for both MACEs and any repeat revascularization. Appendix A shows baseline characteristics between pre-TIMI 0/1 and 2/3 in 1G-DES or 2G-DES groups. Appendix A shows baseline characteristics between pre-TIMI 0/1 and 2/3 groups or between 1G-DES and 2G-DES groups in the total study population. Subgroup analyses for MACEs and any repeat revascularization are shown in Appendix A. In the pre-TIMI 0/1 group, 2G-DES may be preferred to 1G-DES to reduce MACEs and any repeat revascularization in patients who did not experience cardiogenic shock or who did not receive CPR on admission. In the pre-TIMI 2/3 group, 2G-DES may be beneficial to the patients who did not receive CPR on admission, who received culprit-only PCI, or who received more than 3 mm sized DES in reducing MACEs and any repeat revascularization compared with 1G-DES in this study.

## 4. Discussion

In this retrospective cohort study, we compared the two-year major clinical outcomes of the 1G-DES and 2G-DES groups using pre-TIMI. The main findings were as follows. First, in the total study population, the cumulative incidence of MACEs, all-cause death, and CD were significantly higher in the pre-TIMI 0/1 group (Groups A1 and B1) than in the pre-TIMI 2/3 group (Groups A2 and B2). Second, the cumulative incidence of MACEs and any repeat revascularization were significantly higher in the 1G-DES group than in the 2G-DES group among the total study population and in both the individual groups (pre-TIMI 0/1 or 2/3). Third, however, sirolimus-eluting stent showed similar cumulative incidence of any repeat revascularization compared with zotarolimus-eluting stent and biolimus-eluting stent in both pre-TIMI 0/1 and 2/3 groups. Fourth, the cumulative incidence of all-cause death, re-MI, and ST were similar between the 1G-DES and 2G-DES groups.

Stone et al. [18] and De Luca et al. [19] demonstrated that pre-PCI TIMI flow grade 3 is an independent predictor of mortality in AMI. Moreover, Brodie et al. [20] showed that procedural success rates were higher in patients with a pre-TIMI 2/3 than in those with a pre-TIMI 0/1 (97.4% vs. 93.8%, *p* = 0.02). The main advantages of early IRA patency are related with easier guidewire passage and a smaller amount thrombus burden with a relatively lower risk of distal embolization [5]. Others include decreased myocardial infarct size, serious arrhythmic cardiac events, and in-hospital death [21]. In the total study population, our results showing the better major clinical outcomes in the pre-TIMI 2/3 group regarding the cumulative incidences of MACEs, all-cause death, and CD were consistent with previous findings [18,22].

During the two-year follow-up period, in both the individual groups (pre-TIMI 0/1 or 2/3), the cumulative incidence of any repeat revascularization was higher in the 1G-DES groups than in the 2G-DES groups and the increased incidence of any repeat revascularization was associated with an increased incidence of MACEs in the 1G-DES in our study (Table 2 and Table 3). Based on our results, 2G-DES showed sustained better clinical outcomes than 1G-DES with respect to any repeat revascularization regardless of pre-PCI TIMI flow grade. Therefore, although pre-TIMI is an independent predictor of survival in patients with STEMI, it is not significantly related to the major clinical outcomes between 1G-DES and 2G-DES after successful PCI. 2G-DES is using the newer metallic alloys (such as cobalt-chromium and platinum-chromium), allowing thinner strut stent platforms and use of new drug carriers offering reduced incidence of revascularization [1,2,3]. De Luca et al. [23] reported that everolimus-eluting stent was associated with a lower TVR (14.2% vs. 20.1%; HR: 0.63; 95% CI: 0.42–0.96; *p* = 0.03) than the 1G-DES. More recently, Kim et al. [15] reported that the any repeat revascularization rate was significantly higher in the 1G-DES group than in the 2G-DES group in both culprit-only PCI (aHR: 1.345; 95% CI: 1.145–1.705; *p* = 0.001) and multivessel PCI (aHR: 2.444; 95% CI: 1.549–3.855; *p* < 0.001) in their 7266 STEMI patients. Single-center registry data [24] also showed that the five-year cumulative incidence of any repeat revascularization was significantly higher in the 1G-DES group than in the 2G-DES group (16.8% vs. 9.1%, *p* = 0.018) in their 1016 AMI patients. However, in the three-year results of the XAMI (Xience V stent vs. Cypher Stent in Primary PCI for Acute Myocardial Infarction) trial [2], major clinical outcomes were similar between 1G-DES and 2G-DES. Wu et al. [25] showed that the 2G-DES failed to demonstrate a superiority over the 1G-DES in reducing the incidence of TLR (relative risk (RR): 1.73; 95% CI: 0.83–3.64; *p* = 0.15), MACEs (RR: 0.97; *p* = 0.90), or all-cause death (RR: 1.00; *p* = 1.0) in their meta-analysis. It was unclear whether the clinical outcomes of the 2G-DES and the 1G-DES are identical in AMI settings with higher possible hypercoagulable coronary lesions [2]. Otsuka et al. [26] reported that the frequency of neoatherosclerosis were similar between 1G-DES and 2G-DES. A possible explanation for these conflicting results may be related to the different characteristics of the constituents (e.g., STEMI and non-STEMI) in the total study population, differences in follow-up duration, and differences in comparison methods (by individual stent or by group). To clarify our study results, we re-analyzed the cumulative incidence of any repeat revascularization according to types of deployed stents in the total study population, as shown in Table 4 and Figure 3. In this study, PES showed a higher cumulative incidence of any repeat revascularization than SES in the total study population, pre-TIMI 0/1 group, and pre-TIMI 2/3 group. A previous meta-analysis [27] also showed that SES was superior to PES in terms of a significant reduction in the risk of reintervention (HR: 0.74; 95% CI: 0.63–0.87; *p* < 0.001). However, sirolimus-eluting stent showed similar cumulative incidence of any repeat revascularization compared with zotarolimus-eluting stent and biolimus-eluting stent in both pre-TIMI 0/1 and 2/3 groups.

In this study, the cumulative incidence of ST was similar between the 1G-DES and 2G-DES groups in the total or individual groups. Compared to the 1G-DES group, the 2G-DES group included a higher number of patients who received more recently developed and more potent and rapid-acting thienopyridines such as ticagrelor or prasugrel (Table 1). However, the number of these patients was relatively small (<9%) and the prescription rate of clopidogrel (97.3% vs. 83.7%, *p* < 0.001; or 97.7% vs. 84.3%, *p* < 0.001) was significantly higher in the 1G-DES group than in the 2G-DES group. Furthermore, the prescription rate of cilostazole was also higher in the 1G-DES group than in the 2G-DES group (33.7% vs. 17.5%, *p* < 0.001; or 36.1% vs. 17.0%, *p* < 0.001). Therefore, these relatively higher prescription rates of clopidogrel and cilostazole in the 1G-DES group could have induced this similar ST rate between the two groups.

Because a large number of high-volume PCI community and teaching hospitals participated in this multicenter registry analysis, our study may provide impressive information about the comparative benefit of 2G-DES for reducing any revascularization rate compared with 1G-DES in patients with STEMI focused on pre-TIMI.

This study had several limitations. First, the two-year follow-up period in this study was not long enough in order to estimate long-term major clinical outcomes. Moreover, in this study, the temporal range of the inclusion period is large (10 years, 2005–2015) and 1G-DES were probably used several years before 2G-DES (Appendix A). Despite the fact that there is a thorough multivariate adjustment, this situation can lead to two types of uncontrolled biases: (1) Historical bias: General management of patient during the early years might be significantly different that in the late years not only because of invasive or medical treatment but also because or unknown or uncontrolled factors, and (2) during the years where 1G DES and 2G DES coexisted, the operator election of one or other type of device could be influenced by patient’s characteristics. This point is another weak point of this study. Second, because this was a non-randomized retrospective study, there were some underreported and/or missed data. Third, because this study was conducted based on discharge medications and owing to the limitations of registry study, we did not precisely know the adherence or non-adherence of enrolled patients to these drugs during the follow-up period. Fifth, despite multivariate analysis, variables not included in the KAMIR may have affected the study outcomes. Finally, although pre-TIMI is easy and inexpensive, it could be a suboptimal, incomplete measure of myocardial perfusion. Finally, considering that there is a considerable overall mortality, when analyzing single outcomes or composite outcomes not containing all-cause death, a competing risk analysis should be performed. However, in this study, 2.1% (14/676) of the cumulative incidence of any repeat revascularization included all-cause death.

## 5. Conclusions

In conclusion, in this study, 2G-DES showed better clinical outcomes than 1G-DES with respect to MACEs and any repeat revascularization regardless of pre-PCI TIMI flow grade. However, more research is needed to support these results.

## Figures and Tables

**Figure 1 jcm-10-00367-f001:**
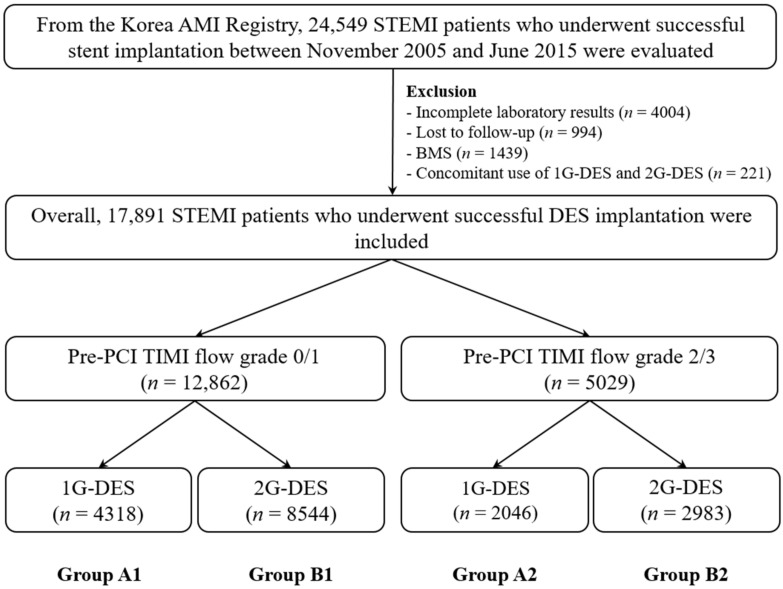
Flowchart. AMI, acute myocardial infarction; STEMI, ST-segment elevation myocardial infarction; BMS, bare-metal stent; 1G-DES, first-generation drug-eluting stent; 2G-DES, second-generation DES; Pre-PCI, pre-percutaneous coronary intervention; TIMI, thrombolysis in myocardial infarction.

**Figure 2 jcm-10-00367-f002:**
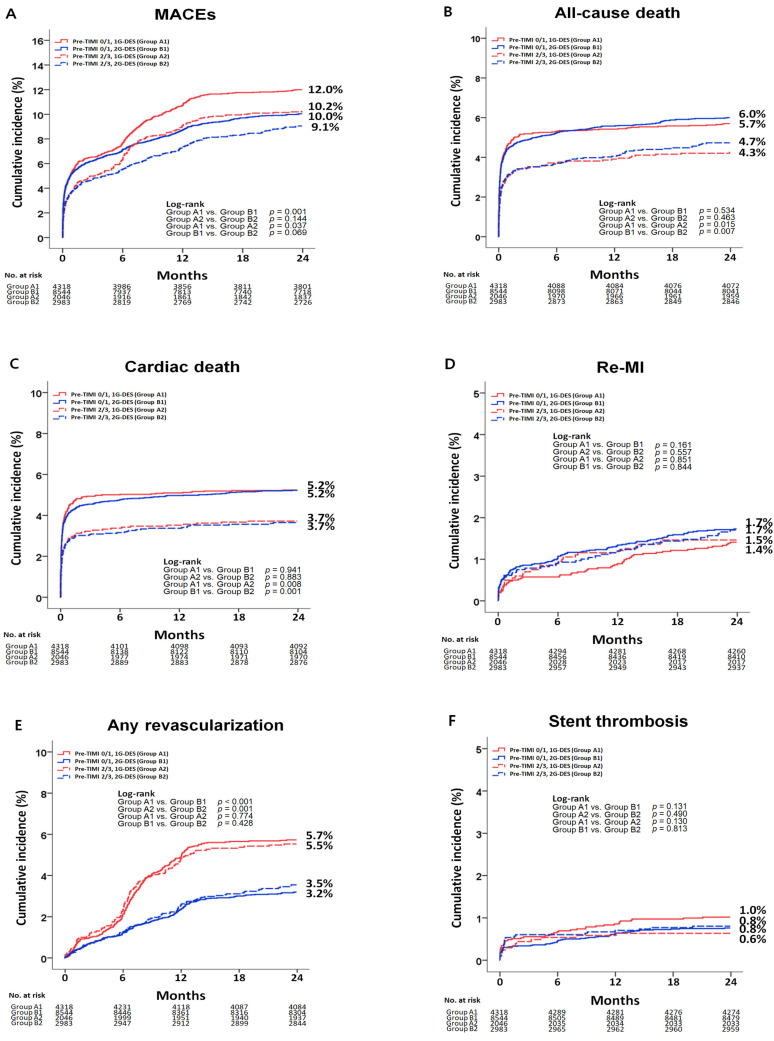
Kaplan–Meier analysis for the MACEs (**A**), all-cause death (**B**), cardiac death (**C**), Re-MI (**D**), any repeat revascularization (**E**), and stent thrombosis (**F**) at 2 years. MACEs, major adverse cardiac events; Re-MI, recurrent myocardial infarction; Pre-TIMI, pre-percutaneous coronary intervention thrombolysis in myocardial infarction flow grade; 1G, first-generation; 2G, second-generation; DES, drug-eluting stent.

**Figure 3 jcm-10-00367-f003:**
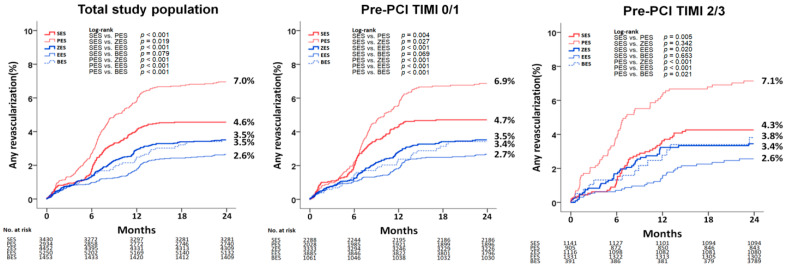
Kaplan–Meier analysis of any repeat revascularization according to the kinds of deployed stents. SES, serolimus-eluting stent; PES, paclitaxel-eluting stent; ZES, zotarolimus-eluting stent; EES, everolimus-eluting stent; BES, biolimus-eluting stent; Pre-PCI, pre-percutaneous coronary intervention; TIMI, thrombolysis in myocardial infarction.

**Table 1 jcm-10-00367-t001:** Baseline characteristics.

Variables	Pre-PCI TIMI 0/1 (*n* = 12,862)			Pre-PCI TIMI 2/3 (*n* = 5029)		
	Group A1 1G-DES (*n* = 4318)	Group B1 2G-DES (*n* = 8554)	*p*-Value	Group A2 1G-DES (*n* = 2046)	Group B2 2G-DES (*n* = 2983)	*p*-Value
Male, *n* (%)	3199 (74.1)	6534 (76.5)	0.003	1533 (74.9)	2270 (76.1)	0.342
Age, years	62.4 ± 12.6	62.3 ± 12.7	0.687	62.6 ± 12.4	63.3 ± 12.8	0.062
LVEF, %	50.2 ± 11.3	50.5 ± 10.8	0.161	51.9 ± 11.8	51.7 ± 11.3	0.539
BMI, kg/m^2^	24.0 ± 3.1	24.2 ± 3.2	0.051	23.8 ± 3.1	23.9 ± 3.2	0.198
Killip classification						
I	3165 (73.3)	6554 (76.7)	<0.001	1570 (76.7)	2431 (81.5)	<0.001
II	624 (14.5)	969 (11.3)	<0.001	259 (12.7)	246 (8.2)	<0.001
III	225 (5.2)	441 (5.2)	0.901	118 (5.8)	161 (5.4)	0.573
IV	305 (7.1)	580 (6.8)	0.561	99 (4.8)	145 (4.9)	0.971
SBP, mmHg	125.2 ± 28.2	126.9 ± 28.6	0.001	128.2 ± 27.2	128.4 ± 27.7	0.785
DBP, mmHg	77.5 ± 17.1	78.0 ± 17.6	0.141	78.7 ± 16.2	78.5 ± 16.2	0.643
Cardiogenic shock, *n* (%)	305 (7.1)	580 (6.8)	0.561	99 (4.8)	145 (4.9)	0.971
CPR on admission, *n* (%)	85 (2.0)	525 (6.1)	<0.001	57 (2.8)	159 (5.3)	<0.001
Hypertension, *n* (%)	1940 (44.9)	3849 (45.0)	0.896	934 (45.7)	1398 (46.9)	0.396
Diabetes mellitus, *n* (%)	1068 (24.7)	2047 (24.0)	0.332	532 (26.0)	792 (26.6)	0.664
Dyslipidemia, *n* (%)	359 (8.3)	922 (10.8)	<0.001	170 (8.3)	322 (10.8)	0.004
Previous MI, *n* (%)	109 (2.5)	259 (3.0)	0.103	42 (2.1)	88 (3.0)	0.057
Previous PCI, *n* (%)	143 (3.3)	387 (4.5)	0.001	66 (3.2)	142 (4.8)	0.008
Previous CABG, *n* (%)	15 (0.3)	27 (0.3)	0.768	5 (0.2)	6 (0.2)	0.766
Previous CVA, *n* (%)	237 (5.5)	415 (4.9)	0.123	102 (5.0)	176 (5.9)	0.163
Previous HF, *n* (%)	40 (0.9)	70 (0.8)	0.533	16 (0.8)	25 (0.8)	0.874
Current smokers, *n* (%)	2064 (47.8)	4004 (46.9)	0.315	987 (48.2)	1364 (45.7)	0.079
Peak CK-MB, mg/dL	209.9 ± 282.3	191.0 ± 244.1	<0.001	138.8 ± 244.9	139.6 ± 225.6	0.915
Peak Troponin-I, ng/mL	64.2 ± 111.1	77.1 ± 294.1	0.001	46.6 ± 68.6	48.2 ± 98.7	0.508
Blood glucose, mg/dL	176.3 ± 80.0	177.4 ± 78.4	0.470	171.5 ± 79.0	173.1 ± 79.0	0.472
NT-ProBNP (pg/mL)	1841.9 ± 3595.0	1638.1 ± 4993.8	0.008	1879.2 ± 3584.3	1884.9 ± 4338.3	0.959
High-sensitivity CRP (mg/dL)	13.4 ± 57.8	9.1 ± 37.4	<0.001	10.5 ± 79.3	7.2 ± 26.9	0.068
Serum creatinine (mg/L)	1.12 ± 1.13	1.08 ± 1.17	0.035	1.16 ± 1.66	1.06 ± 0.90	0.011
Total cholesterol, mg/dL	184.9 ± 44.6	184.7 ± 45.0	0.877	184.4 ± 42.8	181.1 ± 44.1	0.010
Triglyceride, mg/L	127.3 ± 110.5	137.4 ± 112.0	<0.001	125.0 ± 102.9	133.6 ± 109.9	0.006
HDL cholesterol, mg/L	45.0 ± 20.0	43.5 ± 15.8	<0.001	46.1 ± 29.8	43.2 ± 15.2	<0.001
LDL cholesterol, mg/L	118.1 ± 38.4	116.6 ± 39.1	0.059	119.1 ± 44.2	113.5 ± 42.5	<0.001
Discharge medications, *n* (%)						
Aspirin, *n* (%)	4192 (97.1)	8237 (96.3)	0.537	1987 (97.1)	2833 (95.0)	0.152
Clopidogrel, *n* (%)	4202 (97.3)	7154 (83.7)	<0.001	1998 (97.7)	2514 (84.3)	<0.001
Ticagrelor, *n* (%)	6 (0.1)	687 (8.0)	<0.001	5 (0.2)	227 (7.6)	<0.001
Prasugrel, *n* (%)	6 (0.1)	409 (4.8)	<0.001	2 (0.1)	149 (5.0)	<0.001
Cilostazole, *n* (%)	1456 (33.7)	1493 (17.5)	<0.001	739 (36.1)	508 (17.0)	<0.001
Beta-blockers, *n* (%)	3004 (69.6)	6829 (79.4)	<0.001	1508 (73.7)	2429 (81.4)	<0.001
ACEIs/ARBs, *n* (%)	3281 (76.0)	6528 (76.4)	0.597	1632 (79.8)	2353 (78.9)	0.447
CCBs, *n* (%)	266 (6.2)	250 (2.9)	<0.001	162 (7.9)	149 (5.0)	<0.001
Lipid-lowering agents	3887 (90.0)	7613 (89.0)	0.879	1811 (88.5)	2639 (88.5)	0.977
IRA						
Left main, *n* (%)	41 (0.9)	82 (1.0)	0.996	46 (2.2)	81 (2.7)	0.315
LAD, *n* (%)	2144 (49.7)	4271 (50.0)	0.719	1275 (62.3)	1726 (57.9)	0.002
LCx, *n* (%)	450 (10.4)	773 (9.0)	0.012	174 (8.5)	252 (8.4)	0.944
RCA, *n* (%)	1683 (39.0)	3418 (40.0)	0.260	551 (26.9)	924 (31.0)	0.002
Treated vessel						
Left main, *n* (%)	60 (1.4)	120 (1.4)	0.997	71 (3.5)	113 (3.8)	0.593
LAD, *n* (%)	2399 (55.6)	4836 (56.6)	0.260	1399 (68.4)	1951 (65.4)	0.028
LCx, *n* (%)	698 (16.2)	1283 (15.0)	0.092	349 (17.1)	527 (17.7)	0.576
RCA, *n* (%)	1856 (43.0)	3757 (44.0)	0.285	681 (33.3)	1102 (36.9)	0.008
Extent of CAD						
Single-vessel disease, *n* (%)	2055 (47.6)	4737 (55.4)	<0.001	956 (46.7)	1523 (51.1)	0.003
Two-vessel disease, *n* (%)	1339 (31.0)	2437 (28.5)	0.003	643 (31.4)	891 (29.9)	0.238
≥Three-vessel disease, *n* (%)	924 (21.4)	1370 (16.0)	<0.001	447 (21.8)	569 (19.1)	0.016
ACC/AHA lesion type						
Type B1, *n* (%)	602 (13.9)	1075 (12.6)	0.031	444 (21.7)	521 (17.5)	<0.001
Type B2, *n* (%)	921 (21.3)	2420 (28.3)	<0.001	627 (30.6)	1188 (39.8)	<0.001
Type C, *n* (%)	2353 (54.5)	4427 (51.8)	0.004	794 (38.8)	1051 (35.2)	0.010
In-hospital GP IIb/IIIa inhibitor	618 (14.3)	1862 (21.8)	<0.001	156 (7.6)	447 (15.0)	<0.001
IVUS, *n* (%)	267 (6.2)	1397 (16.4)	<0.001	144 (7.0)	623 (20.9)	<0.001
OCT, *n* (%)	0 (0.0)	21 (0.2)	0.001	0 (0.0)	12 (0.4)	0.002
FFR, *n* (%)	4 (0.1)	65 (0.8)	<0.001	2 (0.1)	34 (1.1)	<0.001
Drug-eluting stents						
SES, *n* (%)	2289 (47.0)			1141 (55.8)		
PES, *n* (%)	2029 (53.0)			905 (44.2)		
ZES, *n* (%)		3336 (39.0)			1116 (37.4)	
EES, *n* (%)		3889 (45.5)			1331 (44.6)	
BES, *n* (%)		1062 (12.4)			391 (13.1)	
Others, *n* (%)^a^		257 (3.0)			145 (4.9)	
Door-to-balloon time	59.5 (45.5–72.0)	60.0 (45.0–76.0)	0.688	60.0 (44.0–75.0)	60.5 (44.0–77.0)	0.574
Culprit-only PCI	1444 (33.4)	4569 (53.5)	<0.001	636 (31.1)	1487 (49.8)	<0.001
Multivessel PCI	2874 (66.6)	3975 (46.5)	<0.001	1410 (68.9)	1496 (50.2)	<0.001
Completeness of PCI						
CR, *n* (%)	1954 (68.0)	2762 (69.5)	0.180	988 (70.1)	1069 (71.5)	0.414
IR, *n* (%)	920 (32.0)	1,213 (30.5)	0.180	422 (29.9)	427 (28.5)	0.414
Stent diameter (mm)	3.15 ± 0.38	3.19 ± 0.42	<0.001	3.15 ± 0.37	3.19 ± 0.42	0.002
Stent length (mm)	26.4 ± 6.7	26.6 ± 10.1	0.148	25.4 ± 6.8	25.5 ± 9.9	0.779
Number of stents	1.42 ± 0.74	1.36 ± 0.68	<0.001	1.50 ± 0.80	1.42 ± 0.73	<0.001

Values are means ± standard deviation (SD) or numbers and percentages or as median (quartiles 1–3). The *p*-values for continuous data were obtained from the unpaired *t*-test. The *p*-values for categorical data from chi-square or Fisher’s exact test. 1G-DES, first-generation drug-eluting stent; 2G-DES, second-generation DES; Pre-PCI, pre-percutaneous coronary intervention; TIMI, thrombolysis in myocardial infarction; LVEF, left ventricular ejection fraction; BMI, body mass index; SBP, systolic blood pressure; DBP, diastolic blood pressure; STEMI, ST-segment elevation myocardial infarction, NSTEMI, non-STEMI; CPR, cardiopulmonary resuscitation; MI, myocardial infarction; PCI, percutaneous coronary intervention; CABG, coronary artery bypass graft; HF, heart failure; CVA, cerebrovascular accidents; CK-MB, creatine kinase myocardial band; NT-ProBNP, N-terminal pro-brain natriuretic peptide; Hs-CRP, high-sensitivity-C-reactive protein; HDL, high-density lipoprotein; LDL, low-density lipoprotein; ACEIs, angiotensin converting enzyme inhibitors; ARBs, angiotensin receptor blockers; CCBs, calcium channel blockers; IRA, infarct-related artery; LAD, left anterior descending coronary artery; LCx, left circumflex coronary artery; RCA, right coronary artery; CAD, coronary artery disease; ACC/AHA, American College of Cardiology/American Heart Association; GP, glycoprotein; IVUS, intravascular ultrasound; OCT, optical coherence tomography; FFR, fractional flow reserve; SES, sirolimus-eluting stent; PES, paclitaxel-eluting stent; ZES, zotarolimus-eluting stent; EES, everolimus-eluting stent; BES, biolimus-eluting stent; CR, complete revascularization; IR, incomplete revascularization.

**Table 2 jcm-10-00367-t002:** Clinical outcomes by Kaplan–Meier analysis and Cox-proportional hazard ratio analysis at 2 years.

Pre-PCI TIMI 0/1								
	Cumulative Events (%)			Unadjusted		Adjusted ^a^		
Outcomes	1G-DES Group A1 (*n* = 4318)	2G-DES Group B1 (*n* = 8544)	Log-Rank	HR (95% CI)	*p*-Value	HR (95% CI)	*p*-Value	*p*-for-Interaction
MACEs	517 (12.0)	826 (10.0)	0.001	1.201 (1.076–1.341)	0.001	1.348 (1.184–1.536)	<0.001	0.124
All-cause death	246 (5.7)	503 (6.0)	0.534	1.050 (0.901–1.222)	0.535	1.014 (0.823–1.250)	0.893	0.535
Cardiac death	226 (5.2)	440 (5.2)	0.941	1.006 (0.857–1.181)	0.941	1.086 (0.867–1.361)	0.474	0.941
Re-MI	58 (1.4)	134 (1.7)	0.161	1.246 (0.915–1.696)	0.162	1.366 (0.968–1.926)	0.076	0.017
Any revascularization	234 (5.7)	240 (3.2)	<0.001	1.832 (1.530–2.193)	<0.001	1.938 (1.605–2.340)	<0.001	0.682
ST (definite or probable)	44 (1.0)	65 (0.8)	0.131	1.341 (0.915–1.967)	0.133	1.145 (0.951–2.106)	0.102	0.133
**Pre-PCI TIMI 2/3**								
	**Cumulative Events (%)**			**Unadjusted**		**Adjusted ^b^**		
**Outcomes**	**1G-DES Group A2 (*n* = 2046)**	**2G-DES Group B2 (*n* = 2983)**	**Log-Rank**	**HR (95% CI)**	***p*** **-Value**	**HR (95% CI)**	***p*** **-Value**	***p*-for-Interaction**
MACEs	209 (10.2)	257 (9.1)	0.144	1.146 (0.954–1.375)	0.144	1.415 (1.142–1.953)	0.002	0.144
All-cause death	87 (4.3)	137 (4.7)	0.463	1.106 (0.845–1.447)	0.464	1.311 (0.920–1.964)	0.133	0.464
Cardiac death	76 (3.7)	107 (3.7)	0.883	1.022 (0.762–1.372)	0.884	1.512 (1.010–2.532)	0.053	0.884
Re-MI	29 (1.5)	46 (1.7)	0.557	1.149 (0.722–1.830)	0.557	1.101 (0.665–1.924)	0.708	0.391
Any revascularization	109 (5.5)	94 (3.5)	0.001	1.605 (1.218–2.115)	0.001	1.674 (1.244–2.352)	0.001	0.275
ST (definite or probable)	13 (0.6)	24 (0.8)	0.490	1.267 (0.645–2.489)	0.492	1.178 (0.585–2.670)	0.646	0.492
**1G-DES**								
	**Cumulative Events (%)**			**Unadjusted**		**Adjusted^c^**		
**Outcomes**	**Pre-PCI TIMI 0/1 Group A1 (*n* = 4318)**	**Pre-PCI TIMI 2/3 Group A2 (*n* = 2046)**	**Log-Rank**	**HR (95% CI)**	***p*** **-Value**	**HR (95% CI)**	***p*** **-Value**	***p*-for-Interaction**
MACEs	517 (12.0)	209 (10.2)	0.037	1.186 (1.010–1.393)	0.038	1.222 (1.026–1.454)	0.024	0.158
All-cause death	246 (5.7)	87 (4.3)	0.015	1.352 (1.059–1.726)	0.016	1.459 (1.101–1.932)	0.008	0.218
Cardiac death	226 (5.2)	76 (3.7)	0.008	1.421 (1.096–1.843)	0.008	1.536 (1.136–2.157)	0.005	0.175
Re-MI	58 (1.4)	29 (1.5)	0.851	1.043 (0.668–1.629)	0.852	1.244 (0.762–2.030)	0.383	0.506
Any revascularization	234 (5.7)	109 (5.5)	0.774	1.034 (0.824–1.298)	0.774	1.090 (0.858–1.386)	0.480	0.774
ST (definite or probable)	44 (1.0)	13 (0.6)	0.130	1.606 (0.865–2.981)	0.134	1.735 (0.914–3.301)	0.092	0.134
2G-DES								
	**Cumulative Events (%)**			**Unadjusted**		**Adjusted ^d^**		
**Outcomes**	**Pre-PCI TIMI 0/1 Group B1 (*n* = 8544)**	**Pre-PCI TIMI 2/3 Group B2 (*n* = 2983)**	**Log-Rank**	**HR (95% CI)**	***p*** **-Value**	**HR (95% CI)**	***p*** **-Value**	***p*-for-Interaction**
MACEs	826 (10.0)	257 (9.1)	0.069	1.138 (0.990–1.310)	0.069	1.242 (1.046–1.484)	0.013	0.089
All-cause death	503 (6.0)	137 (4.7)	0.007	1.296 (1.073–1.565)	0.007	1.510 (1.152–1.980)	0.002	0.103
Cardiac death	440 (5.2)	107 (3.7)	0.001	1.449 (1.173–1.790)	0.001	1.840 (1.329–2.483)	<0.001	0.456
Re-MI	134 (1.7)	46 (1.7)	0.844	1.034 (0.740–1.446)	0.884	1.343 (0.905–1.912)	0.124	0.854
Any revascularization	240 (3.2)	94 (3.5)	0.428	1.101 (0.868–1.398)	0.428	1.015 (0.834–1.315)	0.914	0.428
ST (definite or probable)	65 (0.8)	24 (0.8)	0.813	1.058 (0.663–1.690)	0.813	1.114 (0.676–1.836)	0.672	0.813

HR, hazard ratio; CI, confidence interval; MACEs, major adverse cardiac events; Re-MI, recurrent myocardial infarction; ST, stent thrombosis, Pre-PCI, pre-percutaneous coronary intervention; TIMI, thrombolysis in myocardial infarction; 1G-DES, first-generation drug-eluting stent, 2G-DES, second-generation DES; LVEF, left ventricular ejection fraction; SBP, systolic blood pressure, DBP, diastolic blood pressure; CPR, cardiopulmonary resuscitation; DM, diabetes mellitus; PCI, percutaneous coronary intervention; CVA, cerebrovascular accidents; CK-MB, creatine kinase myocardial band; NT-ProBNP, N-terminal pro-brain natriuretic peptide; hs-CRP, high-sensitivity-C-reactive protein; HDL, high-density lipoprotein; LDL, low-density lipoprotein; ACEIs, angiotensin converting enzyme inhibitors; ARBs, angiotensin receptor blockers; CCB, calcium channel blocker, LM, left main coronary artery; LCx, left circumflex artery; RCA, right coronary artery; IRA, infarct-related artery; ACC/AHA, American College of Cardiology/American Heart Association; GP, glycoprotein. IVUS, intravascular ultrasound; OCT, optical coherence tomography; FFR, fractional flow reserve; SES, sirolimus-eluting stent; PES, paclitaxel-eluting stent; ZES, zotarolimus-eluting stent; EES, everolimus-eluting stent. ^a^ Adjusted by male, Killip class I/II, SBP, CRP on admission, dyslipidemia, previous PCI, peak CK-MB, peak troponin-I, NT-ProBNP, hs-CRP, serum creatinine, triglyceride, HDL-cholesterol, clopidogrel, ticagrelor, prasugrel, cilostazole, beta-blocker, CCB, LCx (IRA), single-vessel disease, two-vessel disease, ≥three-vessel disease, ACC/AHA type B1/B2/C lesions, in-hospital GP IIb/IIIa inhibitor, IVUS, OCT, FFR, culprit-only PCI, stent diameter, and number of stents (Table 1). ^b^ Adjusted by Killip class I/II, CRP on admission, dyslipidemia, previous PCI, serum creatinine, total cholesterol, triglyceride, HDL-cholesterol, LDL-cholesterol, clopidogrel, ticagrelor, prasugrel, cilostazole, beta-blocker, CCB, LAD and RCA (IRA and treated vessel), single-vessel disease, ≥three-vessel disease, ACC/AHA type B1/B2/C lesions, in-hospital GP IIb/IIIa inhibitor, IVUS, OCT, FFR, culprit-only PCI, stent diameter, and number of stents (Table 1). ^c^ Adjusted by LVEF, BMI, Killip class I/IV, SBP, DBP, cardiogenic shock, CRP on admission, peak CK-MB, peak troponin-I, blood glucose, beta-blocker, ACEI/ARB, CCB, LM, LAD, LCx, and RCA (IRA), LM, LAD, and RCA (treated vessel), ACC/AHA type B1/B2/C lesions, in-hospital GP IIb/IIIa inhibitor, stent length, and number of stents (Appendix A). ^d^ Adjusted by age, LVEF, BMI, Killip class I/II/IV, SBP, cardiogenic shock, DM, previous CVA, peak CK-MB, peak troponin-I, blood glucose, NT-ProBNP, hs-CRP, total cholesterol, LDL-cholesterol, ACEI/ARB, CCB, LM, LAD, and RCA (IRA), LM, LAD, LCx, and RCA (treated vessel), single-vessel disease, ≥three-vessel disease, ACC/AHA type B1/B2/C lesions, in-hospital GP IIb/IIIa inhibitor, IVUS, other stents, culprit-only PCI, stent length, and number of stents (Appendix A).

**Table 3 jcm-10-00367-t003:** Clinical outcomes by Kaplan-Meier analysis and Cox-proportional hazard ratio analysis at 2 years.

Total Population (Pre-PCI TIMI 0/1 vs. Pre-PCI TIMI 2/3)							
	Cumulative Events (%)			Unadjusted		Adjusted ^a^	
Outcomes	Pre-PCI TIMI 0/1 Group A1 + B1 (*n* = 12,862)	Pre-PCI TIMI 2/3 Group A2 + B2 (*n* = 5029)	Log-Rank	HR (95% CI)	*p*-Value	HR (95% CI)	*p*-Value
MACEs	1343 (10.7)	466 (9.5)	0.012	1.145 (1.030–1.272)	0.012	1.210 (1.075–1.398)	0.005
All-cause death	749 (5.9)	224 (4.5)	<0.001	1.322 (1.139–1.535)	<0.001	1.350 (1.095–1.699)	0.006
Cardiac death	666 (5.2)	183 (3.7)	<0.001	1.437 (1.220–1.692)	<0.001	1.498 (1.180–1.910)	0.001
Re-MI	192 (1.6)	75 (1.6)	0.883	1.020 (0.781–1.332)	0.883	1.150 (0.860–1.541)	0.327
Any revascularization	474 (4.1)	203 (4.4)	0.409	1.072 (0.909–1.263)	0.409	1.057 (0.885–1.275)	0.432
ST (definite or probable)	109 (0.8)	37 (0.7)	0.457	1.152 (0.793–1.673)	0.457	1.231 (0.841–1.814)	0.241
**Total population (1G-DES vs. 2G-DES)**							
	**Cumulative Events (%)**			**Unadjusted**		**Adjusted ^b^**	
**Outcomes**	**1G-DES Group A1 + A2 (*n* = 6364)**	**2G-DES** **Group B1 + B2** **(*n* = 11,527)**	**Log-Rank**	**HR (95% CI)**	***p*** **-Value**	**HR (95% CI)**	***p-*** **Value**
MACEs	726 (11.4)	1083 (9.8)	0.001	1.176 (1.070–1.292)	0.001	1.233 (1.095–1.388)	<0.001
All-cause death	333 (5.2)	640 (5.7)	0.252	1.080 (0.946–1.233)	0.253	1.086 (0.936–1.298)	0.351
Cardiac death	302 (4.7)	547 (4.8)	0.875	1.011 (0.879–1.164)	0.875	1.045 (0.885–1.260)	0.641
Re-MI	87 (1.4)	180 (1.7)	0.133	1.216 (0.942–1.571)	0.134	1.034 (0.754–1.416)	0.837
Any revascularization	343 (5.7)	334 (3.1)	<0.001	1.764 (1.517–2.051)	<0.001	1.832 (1.528–2.197)	<0.001
ST (definite or probable)	57 (0.9)	89 (0.8)	0.379	1.161 (0.833–1.619)	0.379	1.352 (0.921–1.972)	0.107

^a^ Adjusted by age, LVEF, BMI, Killip class I/II/IV, SBP, DBP, cardiogenic shock, DM, peak CK-MB, peak troponin-I, blood glucose, NT-ProBNP, hs-CRP, total cholesterol, triglyceride, clopidogrel, ticagrelor, cilostazole, beta-blocker, ACEI/ARB, CCB, LM, LAD, LCx, and RCA (IRA and treated vessel), single-vessel disease, ≥three-vessel disease, ACC/AHA type B1/B2/C lesions, in-hospital GP IIb/IIIa inhibitor, IVUS, SES, PES, ZES, EES, other stent, culprit-only PCI, complete revascularization, stent length, and number of stents (Appendix A). ^b^ Adjusted by male, BMI, Killip class I/II, SBP, CPR on admission, dyslipidemia, previous MI, previous PCI peak CK-MB, peak troponin-I, NT-ProBNP, hs-CRP, serum creatinine, triglyceride, HDL-cholesterol, LDL-cholesterol, clopidogrel, ticagrelor, prasugrel, cilostazole, beta-blocker, CCB, LAD (IRA), RCA (IRA and treated vessel), single-vessel disease, two-vessel disease, ≥three-vessel disease, ACC/AHA type B1/B2/C lesions, in-hospital GP IIb/IIIa inhibitor, IVUS, OCT, FFR, culprit-only PCI, stent diameter, and number of stents (Appendix A).

**Table 4 jcm-10-00367-t004:** Clinical outcomes by Kaplan–Meier analysis and Cox-proportional hazard ratio analysis according to types of deployed stents.

Pre-PCI TIMI 0/1							
	Cumulative Events (%)			Unadjusted		Adjusted ^a^	
			Log-Rank	HR (95% CI)	*p*-Value	HR (95% CI)	*p*-Value
SES vs. PES	102 (4.7)	132 (6.9)	0.004	1.464 (1.131–1.896)	0.004	1.374 (1.049–1.798)	0.021 ^d^
SES vs. ZES	102 (4.7)	107 (3.5)	0.027	1.358 (1.035–1.781)	0.027	1.577 (1.184–2.100)	0.002 ^e^
SES vs. EES	102 (4.7)	89 (2.7)	<0.001	1.813 (1.364–2.409)	<0.001	2.441 (1.773–3.358)	<0.001 ^d^
SES vs. BES	102 (4.7)	31 (3.4)	0.069	1.449 (0.969–2.166)	0.071	1.948 (1.220–3.109)	0.005 ^e^
PES vs. ZES	132 (6.9)	107 (3.5)	<0.001	2.006 (1.580–2.547)	<0.001	2.025 (1.557–2.634)	<0.001 ^d^
PES vs. EES	132 (6.9)	89 (2.7)	<0.001	2.637 (2.015–3.450)	<0.001	2.919 (2.141–3.981)	<0.001 ^d^
PES vs. BES	132 (6.9)	31 (3.4)	<0.001	2.094 (1.416–3.097)	<0.001	1.917 (1.208–3.042)	0.006 ^d^
**Pre-PCI TIMI 2/3**							
	**Cumulative Events (%)**			**Unadjusted**		**Adjusted ^b^**	
			**Log-Rank**	**HR (95% CI)**	***p*-Value**	**HR (95% CI)**	***p*-Value**
SES vs. PES	47 (4.3)	62 (7.1)	0.005	1.716 (1.175–2.507)	0.005	1.553 (1.035–2.330)	0.034 ^d^
SES vs. ZES	47 (4.3)	36 (3.4)	0.342	1.234 (0.799–1.905)	0.343	1.189 (0.745–1.896)	0.468
SES vs. EES	47 (4.3)	29 (2.6)	0.020	1.721 (1.083–2.735)	0.022	1.853 (1.117–3.074)	0.017 ^d^
SES vs. BES	47 (4.3)	13 (3.8)	0.653	1.151 (0.623–2.128)	0.653	1.057 (0.516–2.164)	0.879
PES vs. ZES	62 (7.1)	36 (3.4)	<0.001	2.085 (1.433–3.033)	<0.001	1.955 (1.278–2.991)	0.002 ^d^
PES vs. EES	62 (7.1)	29 (2.6)	<0.001	2.977 (1.915–4.628)	<0.001	2.873 (1.754–4.709)	<0.001 ^d^
PES vs. BES	62 (7.1)	13 (3.8)	0.021	1.992 (1.095–3.623)	<0.001	2.007 (1.125–3.949)	<0.001 ^d^
**Total population**							
	**Cumulative Events (%)**			**Unadjusted**		**Adjusted ^c^**	
			**Log-Rank**	**HR (95% CI)**	***p*-Value**	**HR (95% CI)**	***p*-Value**
SES vs. PES	149 (4.6)	194 (7.0)	<0.001	1.541 (1.245–1.908)	<0.001	1.415 (1.132–1.769)	0.002 ^d^
SES vs. ZES	149 (4.6)	143 (3.5)	0.019	1.315 (1.045–1.654)	0.019	1.423 (1.116–1.813)	0.004 ^e^
SES vs. EES	149 (4.6)	118 (2.6)	<0.001	1.774 (1.393–2.258)	<0.001	2.203 (1.684–2.883)	<0.001 ^d^
SES vs. BES	149 (4.6)	44 (3.5)	0.079	1.350 (0.964–1.890)	0.080	1.610 (1.093–2.373)	0.016 ^e^
PES vs. ZES	194 (7.0)	143 (3.5)	<0.001	2.033 (1.663–2.485)	<0.001	1.977 (1.583–2.469)	<0.001 ^d^
PES vs. EES	194 (7.0)	118 (2.6)	<0.001	2.728 (2.170–3.430)	<0.001	2.823 (2.174–3.665)	<0.001 ^d^
PES vs. BES	194 (7.0)	44 (3.5)	<0.001	2.070 (1.492–2.872)	<0.001	2.014 (1.370–2.960)	<0.001 ^d^

HR, hazard ratio; CI, confidence interval; Pre-PCI, pre-percutaneous coronary intervention; TIMI, thrombolysis in myocardial infarction; 1G-DES, first-generation drug-eluting stent, 2G-DES, second-generation DES; SES, sirolimus-eluting stent; PES; paclitaxel-eluting stent; ZES, zotarolimus-eluting stent; EES, everolimus-eluting stent; BES, biolimus-eluting stent; BMI, body mass index; LVEF, left ventricular ejection fraction; SBP, systolic blood pressure, CPR, cardiopulmonary resuscitation; MI, myocardial infarction; PCI, percutaneous coronary intervention; DM, diabetes mellitus; CK-MB, creatine kinase myocardial band; NT-ProBNP, N-terminal pro-brain natriuretic peptide; hs-CRP, high sensitivity-C-reactive protein; HDL, high-density lipoprotein; LDL, low-density lipoprotein; CCB, calcium channel blocker; IRA, infarct-related artery; LAD, left anterior descending coronary artery; RCA, right coronary artery; LM, left main coronary artery; ACC/AHA, American College of Cardiology/American Heart Association; GP, glycoprotein; IVUS, intravascular ultrasound; OCT, optical coherence tomography; FFR, fractional flow reserve; PCI, percutaneous coronary intervention. ^a^ Adjusted by male, age, LVEF, BMI, Killip class I/II/III, SBP, CPR on admission, dyslipidemia, previous MI, previous PCI, peak CK-MB, troponin-I, NT-ProBNP, high-sensitivity CRP, serum creatinine, total cholesterol, triglyceride, HDL-cholesterol, aspirin, clopidogrel, ticagrelor, prasugrel, cilostazole, beta-blocker, CCB, LAD, and RCA (IRA and treated vessel), LM (treated vessel), single-vessel disease, two-vessel disease, ≥three-vessel disease, ACC/AHA type B2/C lesions, in-hospital GP IIb/IIIa, IVUS, OCT, FFR, culprit-only PCI, stent diameter, stent length, and number of stents (Appendix A). ^b^ Adjusted by male, age, LVEF, Killip class I/II, cardiogenic shock, CPR on admission, DM, previous PCI, peak CK-MB, NT-ProBNP, total cholesterol, HDL-cholesterol, LDL-cholesterol, clopidogrel, ticagrelor, prasugrel, cilostazole, beta-blocker, CCB, LAD (IRA), RCA (IRA and treated vessel), single-vessel disease, ≥three-vessel disease, ACC/AHA type B1/B2/C lesions, in-hospital GP IIb/IIIa, IVUS, OCT, FFR, culprit-only PCI, stent diameter, stent length, and number of stents (Appendix A). ^c^ Adjusted by male, age, LVEF, BMI, Killip class I/II/III, cardiogenic shock, CPR on admission, dyslipidemia, previous PCI, peak CK-MB, troponin-I, NT-ProBNP, hs-CRP, serum creatinine, total cholesterol, triglyceride, HDL-cholesterol, LDL-cholesterol, clopidogrel, ticagrelor, prasugrel, cilostazole, beta-blocker, CCB, LAD, and RCA (IRA and treated vessel), LM (treated vessel), single-vessel disease, two-vessel disease, ≥three-vessel disease, ACC/AHA type B1/B2/C lesions, in-hospital GP IIb/IIIa, IVUS, OCT, FFR, culprit-only PCI, stent diameter, stent length, and number of stents (Appendix A). ^d^ Statistically significant association after adjustment for multiple comparisons. ^e^
*p*-value was <0.05 but did not achieve statistical significance after adjustment for multiple comparisons.

## Data Availability

Data is contained within the article or supplementary material.

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
