# Peer review of "Comparison of First- and Second-Generation Drug-Eluting Stents in Patients with ST-Segment Elevation Myocardial Infarction Based on Pre-Percutaneous Coronary Intervention Thrombolysis in Myocardial Infarction Flow Grade"

_jcm, 2021, doi:10.3390/jcm10020367_

Round 1

Reviewer 1 Report

In this paper the authors analyse, in a nation-wide large cohort of STEMI patients, the differences between 1G DES vs 2D DES on two-year clinical outcomes. A subgroup analysis was performed depending on pre-PCI TIMI flow (0/1 vs 2/3). The main conclusions were: 

  1. TIMI 0/1 patients presented a worse prognosis than TIMI 2/3 patients (in terms of MACE, all cause death and cardiac death). 
  2. 1G DES risk of 2-years MACE was similar to 2G DES but 1G DES was associated with an increased risk of MACE and any repeated revascularization. 
  3. In the TIMI 2/3 group, sirolimus eluting stents and zotarolimus eluting stents showed a similar risk of repeated revascularization. 

Comments: 

  1. As the authors sought to test if there are differences between 1G and 2G DES clinical outcomes depending on the TIMI flow sub-group, an interaction analysis should be performed and p-for-interaction values should be reported for each comparison. 

  1. When analysing any repeated revascularization according to stent type, a multiple comparison p-value adjustment should be performed since it is not a pre-specified analysis and 7 different comparisons are done. 

  1. The temporal range of the inclusion period is large (10 years, 2005-2015) and 1G DES were probably used several years before than 2G DES. Despite there is a thorough multivariate adjustment, this situation can lead to two types of uncontrolled biases: 1) Historical bias:  General management of patient during the early years might be significantly different that in the late years not only because of invasive or medical treatment but also because or unknown or uncontrolled factors, and 2) During the years where 1G DES and 2G DES coexisted the operator election of one or other type of device could be influenced by patient’s characteristics. These biases should be reflected as limitations of the study. A temporal (yearly) analysis on trends of 1G or 2G DES utilization should be performed. 

  1. Several variables related with the patient haemodynamic status and infarct extension are reported (blood pressure, heart rate, NT-proBNP, troponin, LVEF…) but not Killip-Kimball class, which is a strong and classic predictor of short and long-term prognosis in STEMI. If available, it should be reported and used in the outcomes analyses.

  1. To determine meaningful variables a p<0.01 was used in the univariate analysis. It is not clear how is this p value calculated or what statistical test was used. 

  1. When determining potential confounding factors, a threshold of p <0.001 in the univariate analysis was used. Usually p<0.2, p<0.1 or p<0.05 (in stricter analyses) thresholds are used. Using this low threshold could lead to ignore potential and significant confounders. 

  1. Despite that the primary endpoint contains all cause death, multiple analyses of other endpoints are done. Considering that there is a considerable overall mortality (about 5.4%), when analysing single outcomes (specially any repeated revascularization) or composite outcomes not containing all cause death a competing risk analysis should be performed. Otherwise, results should be interpreted cautiously and reported as a limitation. 

Author Response

Thank you for reviewer's  efforts in evaluating our original submission. We also thank the reviewer for the helpful and valuable comments, which was very useful to us to improve our manuscript.

Our response to reviewer's comments and recommendations are included in the attached file.

Best regards,

Yong Hoon Kim, M.D., Ph.D.

Reviewer 2 Report

This paper is well-written and scientifically sound.

Table 1 shows some significant differences in pre-PCI characteristics such as CPR on admission, previous PCI, CK-MD, and troponin (as an estimate for the loss of cardiomyocytes), CRP, CKD, blood lipids, etc ...

I would like to see the effects of these parameters on the outcome (mortality, MACE, repeat revascularization). I assume that these data are available in table S3.

Author Response

(The authors gave the same response as above.)

Reviewer 3 Report

This is a good example of nicely written and composed meta-review. Such fundamental works are important when comparative questions regarding outcomes between first-generation and second-generation drug-eluting stents are raised. 

While the proofread of text with minor language corrections (like L42 - check the brackets or L57 - check syntax) are needed.

Best regards.

Author Response

(The authors gave the same response as above.)

Round 2

Reviewer 1 Report

1. The interaction analysis should be done for TIMI flow groups and stent type (TIMI x DES generation) but not for each subgroup. 2. A p-value adjustment should be done taking in account multiple comparisons (Benjamini-Hochberg, Bonferroni methods could be appropriate). The multivariate adjustment could be supplementary. 3. OK. Perfect. 4. OK. Perfect. 5. It is still not clear of what hypothesis contrast come this p. “To determine meaningful variables, all variables with p

Author Response

Thank you very much for reviewing our paper.

Our responses to reviewer's comments and recommendations are in the attatched file.

Thank you again.

Yong Hoon Kim, M.D., Ph.D.
